# Centrosomal Protein 55 (CEP55) Drives Immune Exclusion and Resistance to Immune Checkpoint Inhibitors in Colorectal Cancer

**DOI:** 10.3390/vaccines12010063

**Published:** 2024-01-08

**Authors:** Dechen Wangmo, Travis J. Gates, Xianda Zhao, Ruping Sun, Subbaya Subramanian

**Affiliations:** 1Department of Surgery, University of Minnesota Medical School, Minneapolis, MN 55455, USA; wangm005@umn.edu (D.W.); gates173@umn.edu (T.J.G.); zhaox714@umn.edu (X.Z.); 2Department of Pharmacology, University of Minnesota Medical School, Minneapolis, MN 55455, USA; 3Masonic Cancer Center, University of Minnesota, Minneapolis, MN 55455, USA; ruping@umn.edu; 4Department of Laboratory Medicine & Pathology, University of Minnesota, Minneapolis, MN 55455, USA; 5Center for Immunology, University of Minnesota, Minneapolis, MN 55455, USA

**Keywords:** colorectal cancer, CEP55, immune regulation, T cells, immune checkpoint inhibition

## Abstract

Colorectal cancer (CRC) currently ranks as the third most common cancer in the United States, and its incidence is on the rise, especially among younger individuals. Despite the remarkable success of immune checkpoint inhibitors (ICIs) in various cancers, most CRC patients fail to respond due to intrinsic resistance mechanisms. While microsatellite instability-high phenotypes serve as a reliable positive predictive biomarker for ICI treatment, the majority of CRC patients with microsatellite-stable (MSS) tumors remain ineligible for this therapeutic approach. In this study, we investigated the role of centrosomal protein 55 (CEP55) in shaping the tumor immune microenvironment in CRC. CEP55 is overexpressed in multiple cancer types and was shown to promote tumorigenesis by upregulating the PI3K/AKT pathway. Our data revealed that elevated CEP55 expression in CRC was associated with reduced T cell infiltration, contributing to immune exclusion. As CRC tumors progressed, CEP55 expression increased alongside sequential mutations in crucial driver genes (APC, KRAS, TP53, and SMAD4), indicating its involvement in tumor progression. CEP55 knockout significantly impaired tumor growth in vitro and in vivo, suggesting that CEP55 plays a crucial role in tumorigenesis. Furthermore, the CEP55 knockout increased CD8^+^ T cell infiltration and granzyme B production, indicating improved anti-tumor immunity. Additionally, we observed reduced regulatory T cell infiltration in CEP55 knockout tumors, suggesting diminished immune suppression. Most significantly, CEP55 knockout tumors demonstrated enhanced responsiveness to immune checkpoint inhibition in a clinically relevant orthotopic CRC model. Treatment with anti-PD1 significantly reduced tumor growth in CEP55 knockout tumors compared to control tumors, suggesting that inhibiting CEP55 could improve the efficacy of ICIs. Collectively, our study underscores the crucial role of CEP55 in driving immune exclusion and resistance to ICIs in CRC. Targeting CEP55 emerges as a promising therapeutic strategy to sensitize CRC to immune checkpoint inhibition, thereby improving survival outcomes for CRC patients.

## 1. Introduction

Colorectal cancer is the third most common cancer in the United States, and 1 in 20 Americans is expected to develop CRC during their lifetime [1]. Early screening programs have shown effective results by reducing incidence and death rates in patients over 50, but the incidence rate has steadily risen in patients younger than 50 years of age [2]. Immune checkpoint inhibitors (ICIs) have revolutionized cancer treatments with promising outcomes in melanoma, non-small cell lung, and bladder cancer [3]. Owing to their ability to simultaneously regulate the tumor microenvironment and target the heterogeneous tumor cells with limited adverse effects, ICIs have come to the forefront of cancer treatments. In addition to patient-specific treatment, ICIs can potentiate long-term immune surveillance, thus reducing cancer recurrence [4]. Despite the enormous potential of ICI treatment, not all tumors respond to the treatment. Therefore, studies are being carried out to optimize the use of ICI in patients using different predictive biomarkers. One of the commonly used positive biomarkers in clinics is microsatellite instability-high (MSI-H) or defective DNA mismatch repair (dMMR). In 2020, the FDA approved pembrolizumab (anti-PD1) for the first-line treatment of CRC patients with MSI-H/dMMR tumors [5]. However, most CRC patients (~85%) have microsatellite stable (MSS) tumors and do not respond to ICIs, thus making them ineligible for ICI treatment [6]. The mechanistic understanding of why some patients are intrinsically resistant to ICIs while others respond well remains unclear.

Tumor cell-intrinsic factors shape tumors’ response to ICIs. Studies have shown that immune infiltration within the primary tumors positively correlates with better prognosis and response to immunotherapy in CRC [7,8]. ICIs preferentially have robust responses from tumors with pre-existing anti-tumor T cell responses. However, most CRC patients lack robust T cell infiltration due to reasons attributable to tumor cell intrinsic or extrinsic factors [9]. Tumor cell-intrinsic factors encompass genetic alterations in cancer cells that modify various oncogenic pathways and mediate immune evasion. Immune evasion is a major mechanism tumor cells exploit to evade immune surveillance and suppress anti-tumor immune responses. These alterations control cell-intrinsic properties by promoting tumor cell growth and survival and driving resistance to T cell-mediated killing therapies such as ICIs [10]. Characterizing these tumor cell-intrinsic factors will enable us to design novel strategies for targeting the resistant subset of CRC patients and improving treatment outcomes.

Here, we identified centrosomal protein 55 (CEP55) as a potential therapeutic target that can sensitize CRC tumors to ICIs. The role of CEP55 in cytokinesis, particularly during the cytokinetic abscission for the mitotic exit, is well established [11]. Typically, CEP55 is expressed during embryogenesis and is silent in adult tissues except in the testis and thymus; however, the overexpression of CEP55 has been reported in multiple cancer types, including breast, non-small cell lung, prostate, and esophageal squamous cell carcinoma [12,13,14,15]. Furthermore, the overexpression of CEP55 has been implicated in promoting genomic instability and spontaneous tumorigenesis in mice [16]. Subsequent studies have reported the role of CEP55 in regulating the PI3K/AKT pathway in hepatocarcinogenesis [17,18]. Studies have used correlation studies to highlight the link between CEP55 overexpression and cancers such as breast and osteosarcoma [9,11]. However, the underlying mechanism by which CEP55 drives tumorigenesis and resistance to ICIs in colorectal cancer is unknown. A previous functional genomic screening identified CEP55 as one of the 182 genes that, if perturbed, could change the sensitivity of the tumors to T cell-mediated killing therapies [10]. Here, we examined the role of CEP55 in shaping the tumor immune microenvironment in CRC. Furthermore, we investigated whether knocking out CEP55 could enhance T cell infiltration in the tumor, thus improving the efficacy of ICI.

## 2. Materials and Methods

### 2.1. Mice

BALB/c mice (Jackson Laboratory), 6–8 weeks old, were used to implant CT26 and CT26-Luc cells. The mice were kept in autoclaved cages with unlimited food and water. Mice were housed in a pathogen-free facility with a 12 h light–12 h dark cycle. The University of Minnesota Institutional Animal Care and Use Committee (IACUC) approved all animal studies.

### 2.2. Cell Lines

Mouse cell lines MC38 and CT26 (CRL-2638), and human cell lines HCT116 (CCL-247), HCT8 (CCL-244), SW480 (CCL-228), DLD-1 (CCL-221), WiDr (CCL-218), Caco-2 (HTB-37), and HT29 (HTB-38) were used. Dr. Nicholas Haining kindly provided MC38. CT26, DLD-1, HCT-116, HT29, SW480, L-WRN (CRL-3276), and 293T cells were purchased from the American Type Culture Collection (ATCC). Dr. Emil Lou kindly provided HCT-8 cell. Dr. Masato Yamamoto kindly provided Caco-2 and WiDr cells. CT26, DLD-1, and HCT-8 cells were cultured in a complete RPMI 1640 medium containing 100 IU/mL penicillin, 10% heat-inactivated fetal bovine serum (FBS) (Thermo Fisher Scientific; Waltham, MA, USA), and 100 µg/mL streptomycin (Invitrogen Life Technologies; Waltham, MA, USA). The same supplements were added to different base media to make complete media. MC38 and SW480 cells were maintained in a complete Dulbecco’s modified Eagle medium (DMEM) (Thermo Fisher Scientific, Waltham, MA, USA). HCT116 and HT29 cells were maintained in complete McCoy’s 5a medium (Thermo Fisher Scientific, Waltham, MA, USA). WiDr and Caco-2 cells were cultured in complete Eagle’s minimum essential medium (Lonza Bioscience; Walkersville, MD, USA). L-WRN cells were maintained in a complete DMEM medium supplemented with 0.5 mg/L G-418 (Thermo Fischer Scientific, Waltham, MA, USA), 0.5 mg/L hygromycin B (Invitrogen, Waltham, MA, USA), 1 mM Sodium Pyruvate (Thermo Fischer Scientific, Waltham, MA, USA), and 1.5 g/L Sodium Bicarbonate (Sigma-Aldrich, Burlington, MA, USA).

### 2.3. Mouse Organoid Culture

Normal colon crypts were harvested from C57BL/6 WT mice, as previously described [19]. Briefly, the colon was isolated and washed with cold PBS until the supernatant had no debris. Then, the tissue was washed with 5 mM EDTA—PBS to loosen the crypts. The crypts were isolated by vigorously triturating with cold PBS 5 times. Three fractions of the supernatant were collected and viewed under the microscope. The fraction with the most intact crypts was pelleted and resuspended in 30 µL domes of Matrigel Matrix (BD Biosciences; San Jose, CA, USA) in 12-well Tissue Culture treated CytoOne plates (USA Scientific; Ocala, FL, USA).

The sh*Apc*, *Kras^G12D^* (*AK*) organoids were kindly provided by Dr. Khashayarsha Khazaie. AKP organoids were generated by introducing *P53* knockout mutation in AK organoids. Dr. Peter Westcott [20] provided AKPS organoids. Normal colon and AK organoids were cultured in a complete media containing a 1:1 ratio of advanced DMEMF/12 (AdDMEM, Thermo Fisher Scientific, Waltham, MA, USA) and conditioned L-WRN cell supernatant, containing 1X penicillin/streptomycin (Thermo Fisher Scientific, Waltham, MA, USA), 1 mM N-acetylcysteine (Sigma-Aldrich), 10 mM HEPES (Thermo Fisher Scientific, Waltham, MA, USA), 1X B27 (Life Technologies, Carlsbad, CA), 1X N2 (Invitrogen, Waltham, MA, USA), 1 mM N-acetylcysteine (Sigma-Aldrich, Burlington, MA, USA), 1X Glutamax (Thermo Fisher Scientific, Waltham, MA, USA), 100 μg/mL Primocin (Invivogen; San Diego, CA, USA), 10 μM SB202190 (Sigma-Aldrich, Burlington, MA, USA) 10 μM Y-27632 (EMD Millipore; Burlington, MA, USA), 50 ng/mL human EGF (Invitrogen, Waltham, MA, USA), and 10 mM Nicotinamide (Sigma-Aldrich; Burlington, MA, USA). The AKP organoids were cultured in the same complete media as normal colon organoids with an additional treatment of 13 μM of Nutlin-3a (Cayman Chemical; Ann Arbor, MI, USA). The AKPS organoids were maintained in minimal media (AdDMEM F/12 supplemented with 1X B27). The organoids were passaged by treating with TrypLE Express at 37 °C for 10 min and plated domes in a new 12-well plate. The medium was replenished every two days.

### 2.4. Cell Line CRISPR/Cas9 Electroporation Using Neon Electroporation

Confluent CT26 and MC38 cells were dissociated into single cells with 0.05% Trypsin and washed with 1X DPBS. Ribonucleoprotein (RNP) complexes were formed by mixing 3 μL (90 pmol) of sgRNA (Synthego, Redwood City, CA, USA) with 0.5 μL of Cas9 (10 pmol) in 7 μL of Resuspension Buffer R (Thermo Fisher Scientific, Waltham, MA, USA) and incubating for 10 min at room temperature. Cells were resuspended in 5 μL of Resuspension Buffer R, added to the RNP mix, and electroporated using the Neon electroporator. Immediately, the cells were transferred to a pre-warmed 6-well plate containing 2 mL of complete media and incubated in a humidified 37 °C, 5% CO_2_ incubator. Then, 72 h post electroporation, the knockout of CEP55 protein was validated using Western blot. The Synthego Gene Knockout Kit was used. The multi-sgRNA used were GAUUUAAUUAAAAGCAAAUG, CACUGCAUUAGAGAAAUUUA, and CCCAGAGGUGAUUUCAUCCA. Instructions were followed as per the manufacturer’s recommendations.

### 2.5. Western Blot

The cells and organoids were treated with lysis buffer containing 1X RIPA buffer (Thermo Fisher Scientific, Waltham, MA, USA), 1X Phosphatase inhibitor, and 1X Protease inhibitor (Thermo Fisher Scientific, Waltham, MA, USA) on ice for 20 min. The solution was centrifuged at 1200× *g* for 5 min at 4 °C, and the supernatant was aliquoted. Total protein was quantified by using the BCA Protein Assay Kit (Thermo Fisher Scientific, Waltham, MA, USA), and 40 µg was loaded per well for SDS gel electrophoresis. The proteins were separated at 120 V for 1 h and then transferred to a PVD membrane (Thermo Fisher Scientific, Waltham, MA, USA) at 15 V for 30 min. The membrane was imaged for total protein stain using Licor Revert Total Protein Stain. Next, the membrane was washed with TBST, blocked with 5% Bovine Serum Albumin (BSA) in TBST and incubated in primary antibodies overnight at 4 °C. The following day, the membrane was washed with TBST and incubated in a secondary antibody on a shaker for 1 h at room temperature. The membrane was imaged using the Amersham^TM^ ECL Select (Cytiva; Marlborough, MA, USA). The primary antibodies were at 1:125 of CEP55 Polyclonal antibody (Santa Cruz; Dallas, TX, USA) and 1:1500 of Beta-Actin (Cell Signaling Technology, Danvers, MA, USA).

### 2.6. Endoscopy-Guided Injections of Organoids and Cell Lines

AKPS organoids were dissociated into single cells two days before injection and replated. On the day of the injection, organoids were washed in PBS and incubated in Dispase (Stem Cell, Vancouver, Canada) to dissolve the Matrigel for 10 min at 37 °C. Organoids were washed in PBS and passed through a strainer to ensure that no big chunks of organoids would block the needle. Organoids were resuspended in 10% Matrigel and 90% complete media. Mice were anesthetized using isoflurane (4% for induction and 2% for maintenance). Using the Mainz Coloview mini-endoscopic system (Karl Storz Endoscope, Tuttlingen, Germany), 5000 organoids/50 μL were injected per mouse through the channel of the colonoscope and inserted into the colonic mucosa at around 30–40°. The same procedure was followed for the CT26-Luc cells, except instead of incubating in Dispase, the confluent cells were trypsinized and counted, and resuspended in 50% Matrigel and 50% complete media and injected using the Karl Storz Endoscope.

### 2.7. Subcutaneous Tumor Implantation

To establish a subcutaneous model, 2 × 10^5^ CT26 cells were resuspended in 100 μL Matrigel and injected into the right flank of Balb/c mice. Post injection, tumor growth was monitored and measured twice a week, and tumor volume was calculated using the formula: (*length × width*^2^)/2

### 2.8. Treatment Arms

On days 10, 14 and 19, mice injected with tumor cells orthotopically were treated with either 10 mg/kg of anti-PD1 (BioXell, Lebanon, NH, USA) or 10 mg/kg of Rat IgG2b isotype control (BioXell, Lebanon, NH, USA). All the injections were carried out intraperitoneally.

### 2.9. In Vivo Imaging of Mice

The IVIS Spectrum in vivo imaging system (PerkinElmer, Waltham, MA, USA) was used to monitor orthotopic tumor growth. Mice were injected intraperitoneally with 150 mg/kg of D-luciferin (GoldBio, St Louis, MO, USA) 10 min before imaging. Mice were anesthetized with isofluorane. The exposure time was set to 60 s, and a 540 nm filter was used for signal collection.

### 2.10. Immunofluorescence and Histology

After fixing the tissues in 10% formalin followed by paraffin embedding, the FFPE tissue sections were washed with xylene (3 times) and rehydrated with a gradient of ethanol (twice in 100%, 90%, 80%, and 70%). The tissue sections were heated in antigen retrieval buffer (AR9) in a boiling water bath for 12 min for antigen retrieval and then incubated in blocking buffer (5% BSA in PBS with 0.4% Triton-X (PBST)) for 30 min. Slides were incubated in primary antibodies: 1:30 Cep55 (Santa Cruz, Santa Cruz, CA, USA), 1:200 CD8 (Abcam, Cambridge, UK), 1:100 CD4 (Novus Biologicals, Centennial CO, USA), 1:50 FOXP3 (Biotechne), 1:50 TCF1/7 (Cell Signaling, Danvers, MA, USA), Granzyme B (Abcam, Cambridge, UK), 1:100 PD-L1(R&D Systems, Minneapolis, MN, USA), and 1:200 CD3 (Abcam, Cambridge, UK) overnight at 4 °C. The following day, the slides were washed with PBST and incubated in secondary antibodies conjugated with fluorescence for 1 h at room temperature. The slides were washed, mounted with Prolong gold antifade mountant with DAPI, and imaged.

### 2.11. Flow Cytometry

Tumor tissues were harvested and digested in a digestion buffer comprising collagenase (0.5 mg/mL), DNase (50 units/mL), and EDTA (1 mM). Tissues were minced using gentleMACS Dissociator (Miltenyi Biotec, Waltham, MA, USA) for 2 min. The minced tissues were passed through a 70 µM cell strainer and centrifuged at 1200× *g* for 5 min. Red blood cells were removed by treating the cells with red blood cell lysis buffer. Diluted zombie green viability dye (Biolegend, San Diego, CA, USA) (1:1000 in PBS) was used to differentiate live and dead cells. Cells were stained with cell surface marker antibody cocktails and intracellular marker antibody cocktails and fixed using the cyto-fast fix perm buffer set and true nuclear or transcription factor buffer set (Biolegend, San Diego, CA, USA). The data were analyzed with FlowJo software, ver 10.

### 2.12. Isolation of Extracellular Vesicles

The cells were cultured in complete media, and 1 day before the isolation, complete media was replaced with a media supplemented with 10% exosome-depleted FBS (Gibco, Waltham, MA, USA). The detailed protocol had been previously described [21]. Briefly, the cell culture supernatant was centrifuged at 300× *g* for 10 min to remove cells. The supernatant was centrifuged at 2000× *g* for another 10 min to remove cell debris. The pellet was discarded, and the supernatant was centrifuged at 10,000× *g* for 30 min at 4 °C. The pellet was harvested and suspended in PBS and centrifuged in an ultracentrifuge machine at 100,000× *g* for 70 min at 4 °C. The ultracentrifugation was repeated once again. The pellet was resuspended in the PBS.

### 2.13. Statistical Analysis

GraphPad Prism software (Version 8) was used to graph and perform all statistical analyses. Data are shown as means ± SEMs. An unpaired student’s *t*-test was used to compare the results between the two groups. One-way analysis of variance (ANOVA) was performed to compare multiple groups, followed by Bonferroni correction. Contingency tables were analyzed by the Chi-square test. A two-tail *p* value of less than 0.05 was deemed statistically significant.

## 3. Results

### 3.1. Overexpression of CEP55 in CRC Reduces Immune Cell Infiltration

To evaluate the expression levels and immunoregulatory function of CEP55 in CRC, we examined the expression of CEP55 in the TCGA dataset. First, we analyzed the 182-core gene set identified through the use of a CRISPR screen by Lawson et al.; the alterations in these genes affected the sensitivity of tumor cells to cytotoxic T cell mediated killing [10]. Our analysis revealed that CEP55 was upregulated along with twenty other genes in the TCGA-CRC dataset (Figure 1A). Interestingly, CEP55 expression was also elevated in twelve different cancer types, including colorectal, lung, breast, liver hepatocellular, kidney renal clear cell, kidney renal papillary cell, bladder urothelial, uterine corpus endometrial, stomach, thyroid, and prostate cancers (Figure 1B). Notably, CEP55 expression was higher in CRC tissues than in the normal colon tissues. To confirm this, we performed CEP55 gene expression analysis using an independent cohort of 22 CRC patients along with patient-matched normal colon tissues from the University of Minnesota Cancer Center (Figure 1C).

Next, we analyzed the tumor immune infiltration in the TCGA-COAD and READ datasets using the CIBERSORT algorithm. Tumors with higher levels of CEP55 expression in CRC had reduced CD4^+^ T cells relative fraction immune cell infiltration, but no significant difference was observed in CD8^+^ T cells infiltration (Figure 1D,E). Lawson et al. identified CEP55 as one of the 182 core conserved sets of genes mediating cancer-intrinsic cytotoxic lymphocyte immune evasion [10]. CEP55 was determined to be one of the critical chromosomal instability signatures in breast, ovarian, and small-cell lung cancer [22]. Additionally, in multiple myeloma, CEP55 was implicated in the top 10 genes promoting drug resistance [23]. Chromosomal instability caused by genes such as CEP55 can cause loss of neoantigen and, therefore, can contribute to immune evasion [24]. Altogether, our integrative TCGA data analysis shows that CEP55 expression is consistently elevated in multiple cancer types, and overexpression of CEP55 is negatively associated with immune cell infiltration.

### 3.2. CEP55 Expression Increases with the Addition of Sequential Driver Gene Mutation in CRC

CRC progression is driven by the sequential acquisition of set mutations in *APC*, *KRAS*, *TP53*, and *SMAD4*, a process known as the adenoma-carcinoma sequence [25,26,27]. To investigate how the expression of CEP55 changes with tumor progression, we used CRC organoids harboring different combinations of CRC driver gene mutations. The AKPS tumor organoid (tumoroid) with mutations in *APC*, *KRAS*, *TP53*, and *SMAD4 genes* presented key features of advanced human CRC, such as invasive cancer, liver metastasis, and lymph node metastasis (Figure 2A–F). Furthermore, as the organoids gained new mutations sequentially, the normal colon morphology changed from normal tissue to serrated adenoma with dysplasia and finally to adenocarcinoma (Figure 2G). First, we compared CEP55 expression in organoids with mutations in (i) sh*Apc*, *Kras mutants* (AK), (ii) shApc, *Kras*, and *p53 mutants* (AKP), and (iii) shApc, *Kras*, *p53*, and *Smad4 mutants* (AKPS).

CEP55 mRNA expression increased with the addition of new mutations in the organoids; although there was no significant difference between shApc and AK organoids, the addition of *Smad4* significantly increased CEP55 expression when compared to both AK and shApc (Figure 2H). Notably, CEP55 expression increased with each new mutation in organoid-derived tumor tissues, with the basal level expression in shApc-derived polyps, followed by AKP tumors, and the highest in AKPS tumors (Figure 2I). We also evaluated CEP55 expression in other cancer cell lines besides organoids. Both mouse cell lines CT26 and MC38 and a panel of human CRC cell lines showed elevated levels of CEP55 (Figure 2J). These results suggested that CEP55 expression increases with colorectal cancer progression.

### 3.3. Knockout of CEP55 in Cancer Cells Impairs Tumor Growth

CEP55 overexpression has been reported to cause accelerated *P53*^+/−^ induced tumorigenesis in mice [16]. Initially, the role of CEP55 was described only in the mechanism of cytokinesis, but recent studies have suggested its involvement in tumor progression [28]. Although previous studies have shown the association between CEP55 and cancer progression, the mechanisms of CEP55’s role in CRC are not fully understood. To investigate whether the knockout of CEP55 affects tumor development in vivo, we knocked out the expression of CEP55 in CT26 and MC38 cell lines using CRISPR-Cas9 genomic editing technology (Figure 3A). CEP55 is a potential exosomal marker in head and neck squamous cell carcinoma [29,30]. However, we did not detect CEP55 protein in mouse CT26 cell line-derived exosomes (Figure 3B). The primary role of CEP55 is regulating cytokinesis [31], but Tedeschi et al. showed that CEP55 is mainly dispensable for cell division in embryogenesis [31]. To determine if the knockout of CEP55 is sufficient to reduce cancer cell proliferation rate, we tested the cell proliferation ability of CT26 with and without CEP55 expression. The knockout of CEP55 in CT26 significantly reduced cell proliferation in vitro (Figure 3C). Subsequently, we injected the CEP55KO CT26 cells subcutaneously into naïve Balb/c mice and monitored tumor growth for 21 days (Figure 3D,E). Notably, knocking out CEP55 significantly slowed tumor growth compared to wild-type (WT) and vector control cells in animal models (Figure 3E). Consistent with the in vitro data and previous studies on other cancer types, the knockout of CEP55 in CRC impaired tumor progression and reduced tumor size.

### 3.4. Elevated CEP55 Expression in CRC Drives Immune Cell Exclusion and T Cell Dysfunction

Our findings suggested that CEP55 overexpression drives tumorigenesis. Therefore, it is pertinent to investigate the underlying mechanism of anti-tumor immunity. We analyzed the immune landscape in the tumors derived from CEP55 KO cell lines. Although we did not observe a significant change in the tumor infiltration of CD4^+^ T cells (Figure 4B), it is worth noting that CEP55 KO-derived tumors showed a significantly elevated CD8^+^ T cell infiltration (Figure 4A,C). An essential step in the maturation of functional T cells is effector differentiation, characterized by the production of enzymes such as granzyme B [32]. We detected elevated levels of granzyme B by CD8^+^ T cells in the tumors with knockout CEP55 compared to WT tumors (Figure 4A). Higher production of granzyme B has been associated with lower tumor recurrence and improved survival in CRC patients [33,34]. Next, we sought to interrogate the state of immune suppression in the tumors. It is well-established that immune suppression can also cause resistance to ICIs along with immune exclusion. One major player in facilitating an immune suppressive tumor microenvironment is regulatory T cells (Tregs) [35]. Hence, we analyzed the amount of Forkhead Box P3 (FOXP3^+)^ Treg infiltration. Consistent with the results, knocking down CEP55 reduced Treg infiltration in tumor tissues.

Additionally, tumors derived from knockout CEP55 cells had higher infiltration of progenitor-exhausted T cells (Figure 4H). TCF is typically expressed by naïve and memory T cells. TCF1^+^ T cells have been identified as an early progenitor T cell exhaustion marker in cancer [20,36,37]. Altogether, these results demonstrate that the knockout of CEP55 can rescue T cell dysfunction and improve intratumoral T cell infiltration in CRC.

### 3.5. Loss of CEP55 Enhanced Tumor Response to Immune Checkpoint Inhibitor

The subcutaneous tumor model showed that CEP55 knockout improves tumor T cell infiltration and restores an immune-active microenvironment. One major reason for the FDA approval of anti-PD1 for the treatment of MSI-CRC is the higher likelihood of these patients responding to the treatment due to the pre-existing T cells in the tumor, which is scarce in MSS-CRC [38]. We next asked whether deleting CEP55 can change the tumor’s response to ICIs. Studies concerning testing ICIs should carefully consider using an appropriate tumor model because tumor location has been reported to heavily influence immune responses in CRC animal models. The CRC orthotopic animal model very closely recapitulates the human CRC tumor microenvironment and presents comparable levels of tumor immune infiltration [39]. Therefore, we opted for the orthotopic model, a more clinically relevant model in which tumor cells were injected into the colonic mucosa using endoscopy. Luciferase-expressing CT26 cell line-derived control and CEP55 KO tumors were treated with either anti-PD1 or Isotype IgG2b (control) on days 10, 14, and 19 (Figure 5A,B). The CEP55 knockout tumors were more sensitive to anti-PD1 (Figure 5C,E). Notably, treating the CEP55 KO tumor with anti-PD1 significantly reduced tumor volume compared to the control tumor treated with anti-PD1 and the CEP55 KO tumor treated with IgG2b isotype (Figure 5D,E). Notably, the wild-type tumor treated with anti-PD1 had a tumor size like that of the CEP55 KO tumor treated with isotype (Figure 5E). Despite the delayed tumor growth, we did not observe a statistically significant difference in CD4^+^ and CD8^+^ T cell infiltration. Although no significant difference was detected in the levels of granzyme b, we observed an elevated level in KO tumors treated with anti-PD1. Given the sample size, it is likely that the statistical difference is not observable (Figure 5E–H). We also determined whether the knockout of CEP55 in cancer cells could potentially elevate the expression of PD-L1, thereby enhancing sensitivity to anti-PD1 therapy. Immunofluorescence analysis was conducted on orthotopic tumor tissues from CEP55 knockout and CT26 wild-type (WT) tumors. We did not observe a significant increase in PD-L1 expression in CEP55 knockout tumors compared to CT26 WT tumors (*p* = 0.2998) (Appendix A). This suggests that CEP55 may influence sensitivity to immune checkpoint inhibitors (ICIs) through alternative intrinsic immune regulatory mechanisms. Our results suggested that the treatment of CEP55 knockout tumors with anti-PD1 leads to a reduction in tumor growth and size, suggesting the potential for improved efficacy by inhibiting CEP55. It is crucial to acknowledge that the sample size for these observations is limited, emphasizing the need for future investigations to characterize the synergistic effects of CEP55 knockout tumors more comprehensively. A larger sample size and robust mechanistic validation will be integral in further elucidating the impact of CEP55 inhibition on the efficacy of ICIs in tumors.

## 4. Discussion

Poor response of CRC to immune checkpoint inhibitors is a major unmet clinical need. Different intrinsic and extrinsic factors underlie such immunotherapy resistance. Extensive studies have shown that tumor mutational burden, alterations in signaling pathways, T cell dysfunction, and the evolution of tumor microenvironment into a more immune suppressive niche contribute heavily to poor ICI response and driving therapy resistance [40,41,42]. Specifically in MSS-CRC, low neoantigen load and the lack of immune cell infiltration are key drivers of resistance [20,43]. However, merely using the tumor mutational burden or neoantigen load as a predictive biomarker for ICI treatment would be unsophisticated because CRC, like many MSS tumors, is intrinsically heterogeneous [1]. The strategies to effectively treat CRC patients with ICIs remain challenging and unsuccessful. In addition to the factors mentioned above, insights into the mechanisms of resistance to ICIs are required to improve treatment strategies and better predict patient therapeutic responses.

We identified CEP55 as a tumor cell-intrinsic factor important in mediating a tumor-suppressive microenvironment to promote T cell exclusion. This observation raises the possibility that the function of CEP55 is not confined to cell proliferation and embryogenesis alone, both of which have been extensively studied [31,44,45,46,47,48,49,50]. Consistent with previous studies [12,14,16,28,45,51,52,53,54], we found that CEP55 expression was upregulated in multiple cancer types. More importantly, there was an inverse relationship between CEP55 expression and immune cell infiltration, particularly tumor T cell infiltration. It is widely accepted that a key feature of MSS-CRC is poor T cell infiltration, which is a considerable obstacle considering the majority of CRC patients exhibit this phenotype [55]. These findings suggest an immunoregulatory role of CEP55 in promoting tumorigenesis.

To further probe the role of CEP55 in tumorigenesis, we used tumor organoids as preclinical CRC models. Mouse models and conventional preclinical models have provided invaluable insights into cancer studies. However, none accurately recapitulates the essential features of advanced human CRC due to the difficulty in encompassing tumor heterogeneity. This contributes to the difference in patients’ responses to treatment in clinical trials [56,57]. Our findings showed that CEP55 expression significantly increases as the tumor gains sequential mutations in CRC driver genes, a phenomenon driving adenoma to carcinoma progression of CRC. The tumor becomes more invasive and aggressive with each new mutation, and the CEP55 expression undergoes significant upregulation during tumor progression. This is a critical aspect of this study because it suggests that CEP55 upregulation is not an abrupt change but a gradual process accompanying each phase of tumor growth in CRC. This finding corroborates with a previous study that reported that p53 inactivation in CRC negatively regulates CEP55 expression through Polo-like kinase 1 (PLK1) downregulation [11,28,44,46].

A key question is whether CEP55 has an immunoregulatory role in CRC. Using the CRISPR-Cas9 gene editing system, we generated CEP55 knockout cell lines using the CT26 and MC38 cells. Leveraging the advantages of both the subcutaneous and orthotopic models, the edited tumor cells were injected into mice. In both models, knockout CEP55 tumors grew significantly slower than the WT and control tumors. Furthermore, loss of CEP55 in tumors dramatically rescued the immune suppressive tumor environment by reducing the tumor suppressive regulatory T cells and TCF1 markers. These immune suppressive markers and cells can facilitate a microenvironment that promotes T cell exclusion and dysfunction, leading to acquired resistance to ICIs.

Interestingly, in the subcutaneous model, the CEP55 knockout tumors had higher CD8^+^ T cell infiltration with elevated granzyme b production, but there was no significant change in the CD4^+^ T cell infiltration. This can be attributed to the nature of the subcutaneous model since, in this model, tumor cells are injected into the right flank of the mice. Thus, tumor cells are not subjected to a physiological tumor microenvironment. Although a significant difference in granzyme b production was noted between WT and KO tumors, no statistical difference was detected between the control and KO tumors. It is established that compared to orthotopic models, the subcutaneous model generally has fewer T and B cell infiltration [39].

More importantly, anti-PD1 treated WT showed a tumor size similar to isotype-control treated CEP55 KO tumor. This is important because the knockout CEP55 and anti-PD1 combination significantly reduces tumor size. Additionally, CEP55 knockout alone is sufficient to mimic the effects of anti-PD1 in WT tumors. Future studies entail increasing the sample size, monitoring the tumor growth for extended periods, and examining the mechanism of anti-tumor response in the CEP55 KO tumors. Using the orthotopic model, we asked whether CEP55 can sensitize the tumor to anti-PD1. Indeed, CEP55 knockout tumors respond better to anti-PD1 treatment. These findings showed that therapeutically targeting CEP55 with immune checkpoint inhibitors could be highly efficacious in CRC.

Altogether, the results of this study define a novel immunoregulatory role of CEP55 in CRC. Within the scope of this study, we also looked at CEP55 expression in tumor-secreted extracellular vesicles (TEVs), which are critical for intercellular communications between tumor cells and immune cells [58]. In cancer, these TEVs were shown to suppress the immune system and promote angiogenesis and drug resistance [28,58,59,60]. CEP55 mRNA had been enriched in human CRC cell-line-derived EVs [29], but we did not detect CEP55 protein expression in the mouse CRC cell-line-derived EVs. The CEP55 mRNA expression may undergo post-transcriptional modifications, thus not making itself a part of the TEV cargo. Alternatively, perhaps the function of CEP55 is not directly part of the intracellular communications via TEVs, but rather, in the production of TEVs, CEP55 is known to interact with TGS101 and ALIX. Therefore, it will be essential to investigate whether CEP55 is involved in TEV production. The AKPS organoid model provides an excellent CRC preclinical model and can be genetically engineered to further investigate the resistance mechanisms to ICI mediated by CEP55.

### Limitations of Study

This study highlights a potential regulatory role for CEP55 in the context of CRC. Our initial orthotopic model explores the response of CEP55 knockout tumors to ICIs. While we observed a reduction in endpoint tumor volumes, we acknowledge the small sample size of mice. Future studies are warranted to augment the sample size in order to discern whether the observed tumor volume reduction is biologically significant. Furthermore, inhibitory cytokine production is crucial for inducing T cell exhaustion and Treg accumulation in the tumor environment. However, this study did not assess changes in pro-inflammatory or anti-inflammatory cytokines. Additional mechanistic experiments are required to understand how CEP55 influences the expression and abundance of inhibitory cytokines. We also acknowledge that the conclusions drawn from this study apply exclusively to preclinical mouse models and may not predict similar outcomes in humans.

## 5. Conclusions

In conclusion, our study unveils a potential immunoregulatory role of CEP55 in CRC, illuminating its influence on the tumor microenvironment and therapeutic response to ICIs. Through CRISPR-Cas9 gene editing and preclinical CRC models, we demonstrated that CEP55 knockout leads to slower tumor growth, diminished immune suppressive markers, and a potential enhancement in responsiveness to anti-PD1 treatment. These findings advocate targeting CEP55 in combination with ICIs as a promising therapeutic strategy for CRC. This research deepens our understanding of CRC immunotherapy resistance and introduces a novel target that could significantly improve treatment outcomes. The groundwork laid by our findings sets the stage for future investigations, which will further explore the immunoregulatory role of CEP55 in response to immune checkpoint inhibitors. Leveraging advanced preclinical models in forthcoming studies will provide a more comprehensive understanding of CEP55’s role and implications for refining therapeutic approaches in CRC.

## Figures and Tables

**Figure 1 vaccines-12-00063-f001:**
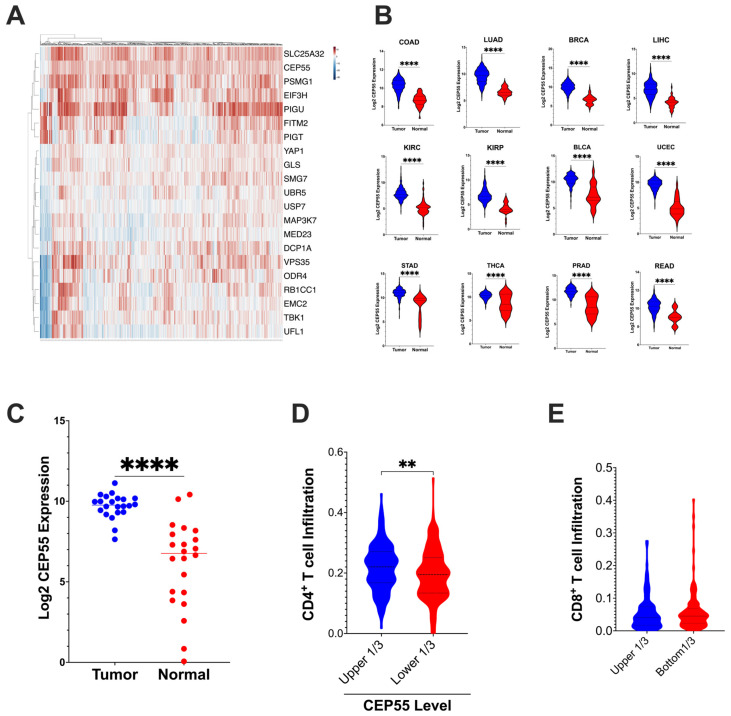
CEP55 expression in tumor tissues and tumor immune cell infiltration (**A**) CEP55 expression, along with twenty other genes identified to regulate the tumor’s sensitivity to T cell-mediated killing 10, is upregulated in CRC. (**B**) CEP55 mRNA expression in the TCGA dataset. Tumor tissues in all the twelve cancer types had higher levels of CEP55 expression than the normal tissues. (**C**) CEP55 mRNA expression in an independent cohort of twenty-two CRC patients. Tumor tissues had an elevated CEP55 expression compared to patient-matched normal colon tissues. (**D**,**E**) CIBERSORT generated the immune profile of the TCGA-CRC dataset. Higher expression of CEP55 had reduced infiltration of CD4^+^ T cells (relative fraction), but no significant difference was detected in CD8^+^ T cells relative fraction infiltration. (The significance of mean differences was calculated using unpaired student *t*-test. ** *p* < 0.01, **** *p* < 0.0001, ns: not significant).

**Figure 2 vaccines-12-00063-f002:**
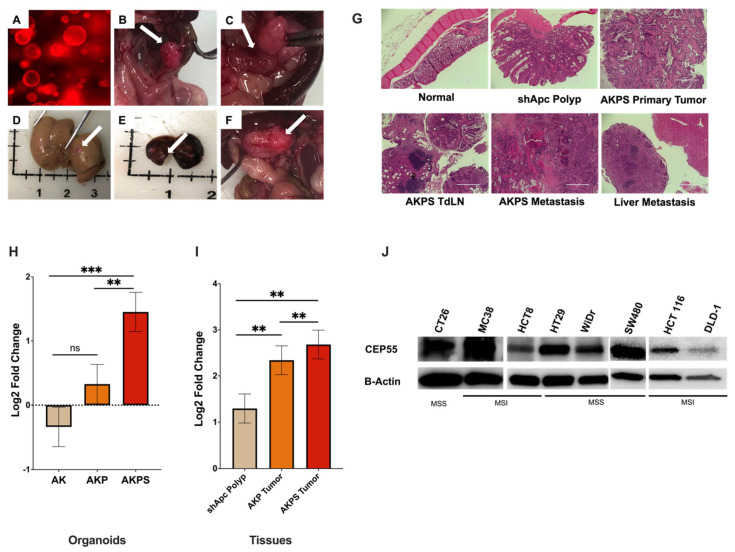
Expression of CEP55 increases with tumor progression. (**A**–**F**) AKPS tumoroids closely recapitulated human CRC pathophysiology. (**A**) AKPS tumoroids tagged with RFP organoids cultured in vitro, (**B**–**F**) endoscopy-guided injection of AKPS tumoroids successfully formed, (**B**) solid primary tumor in the colon, (**C**) lymph node metastasis, (**D**) liver metastasis, (**E**) lung metastasis (**F**) other metastasis, (**G**) H&E staining showed that AKPS tumoroids mimicked advanced human CRC condition. AKPS tumor metastasized to tumor-draining lymph nodes and the liver. (**H**) In organoids, CEP55 expression increased with the addition of mutations in CRC driver genes. (**I**) In organoid-derived tumor tissues, AKPS tumor expressed the highest level of CEP55, followed by AKP tumor and the least by shApc polyp. (**J**) Both mouse and human CRC cell lines with MSS and MSI phenotypes expressed CEP55. Specifically, human MSI-CRC cell lines showed a lower level of CEP55 expression than MSS-CRC cell lines. (For statistical analysis involving more than two groups: the primary *p*-value (topmost) signifies the ANOVA analysis, while subsequent *p*-values represent post hoc analyses between specific pairs of groups. Significance levels are denoted as ** *p* < 0.01, *** *p* < 0.001), ns: not significant.

**Figure 3 vaccines-12-00063-f003:**
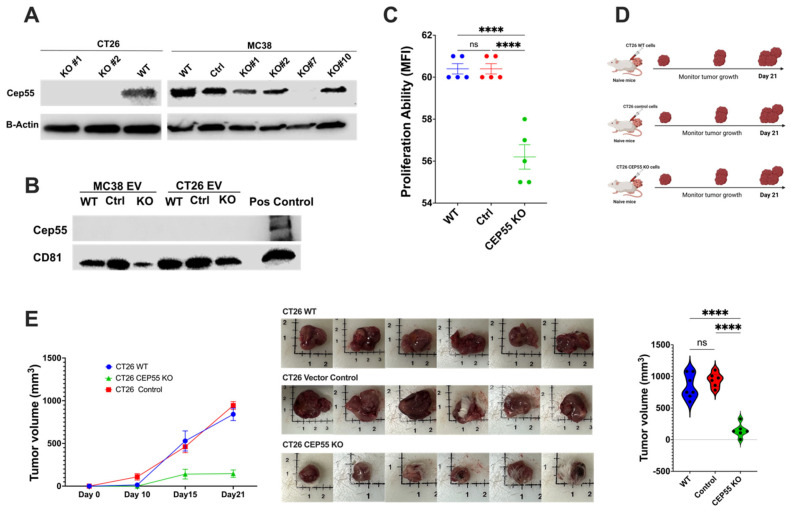
Knockout of CEP55 expression limits tumor progression. (**A**) Western blot analysis of knockout of CEP55 in CT26 and MC38 cell lines. CT26 clone #1 and clone #2 were expanded and used for downstream experiments. MC38 clone #7 was used for further studies. (**B**) CEP55 was not detected in tumor-cell-secreted extracellular vesicles. (**C**) Knockout of CEP55 significantly reduced the cell proliferation ability of the CT26 cell line in vitro. (**D**) CT26 (WT, vector control, CEP55 KO) cells were injected subcutaneously into the right flank of syngeneic mice, and tumor growth was monitored for 21 days. (*n* = 5 per group) (**E**) Tumors with CT26 CEP55 KO cells grew much slower with reduced tumor volume in Balb/c mice. (The one-way ANOVA analysis and Bonferroni correction were used for more than two group statistical analyses. The primary *p*-value denotes the ANOVA analysis, while subsequent *p*-values reflect post hoc analyses conducted between two specific groups. Significance levels are denoted as *****p* < 0.0001, ns: not significant. CD81: Cluster of Differentiation 81 (Tetraspanin marker); WT: Wild Type; Control: Vector Control; CEP55 KO: CEP55 Knockout).

**Figure 4 vaccines-12-00063-f004:**
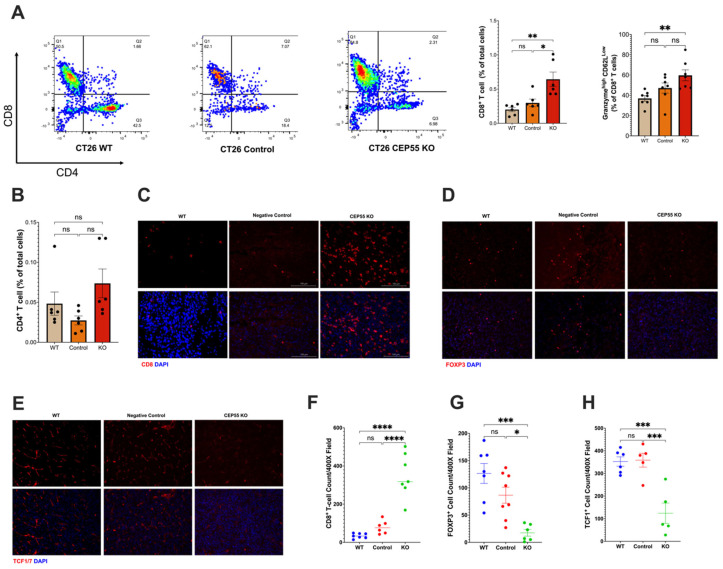
Knockout of CEP55 reverses immune exclusion. (**A**) Knocking out CEP55 expression in tumor cells improved CD8^+^ T cell infiltration and upregulated granzyme B. (**B**) Conversely, no significant change was observed in CD4^+^ T cell infiltration. (**C**,**F**) Immunohistochemistry staining of CD8^+^ T cells confirmed that CEP55 Knockout tumors have a higher frequency of T cells. (**D**,**G**) Infiltration of regulatory T cells was significantly reduced in tumors with knockout CEP55. (**E**,**H**) Early progenitor exhausted T cell frequency was lower in tumor tissues derived from knockout CEP55. (The One-way ANOVA analysis and Bonferroni correction were used for more than two group statistical analyses. The topmost *p*-value corresponds to the ANOVA analysis, while the subsequent *p*-values signify post hoc analyses conducted between specific pairs of groups. Significance levels are denoted as * *p* < 0.05, ** *p* < 0.01, *** *p* < 0.001, *****p* ≤ 0.0001, ns: not significant. CD8: Cluster of Differentiation 8; CD4: Cluster of Differentiation 4; FOXP3: Forkhead Box P3; TCF1: T Cell Factor 1, *n* = 4–5).

**Figure 5 vaccines-12-00063-f005:**
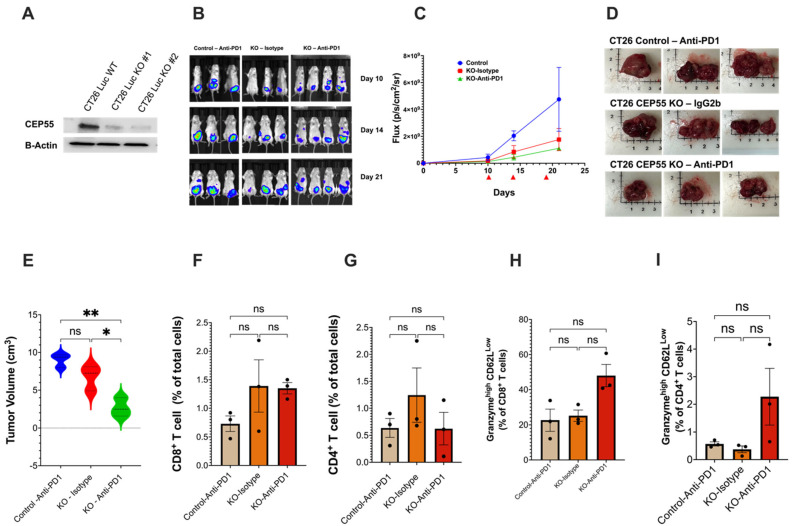
Loss of CEP55 sensitizes the tumor to immune checkpoint inhibitors. (**A**) Generation of CT26-Luc cells with knockout CEP55. (**B**–**E**) Tumor growth was regularly monitored for 21 days using the IVIS spectrum small imaging system by measuring the bioluminescence intensity of the tumor cells implanted in the colon mucosa layer. Anti-PD1 or Isotype IgG2b were injected on days 10, 14, and 19 post-tumor cell injection. Endpoint tumor volume showed significantly reduced tumor size in CEP55 KO tumors treated with Anti-PD1 (*n* = 3–4). (**F**–**I**) FACs analysis of the tumor tissues showed that in the orthotopic model, loss of CEP55 did not improve CD8^+^ and CD4^+^ T cell infiltration. Levels of granzyme b did not show a significant difference. (The One-way ANOVA analysis and Bonferroni correction were used for more than two group statistical analyses. The uppermost *p*-value indicates the ANOVA analysis, and other *p*-values indicate the posthoc analysis between two specific groups. * *p* < 0.05, ** *p* < 0.01, ns: not significant Luc: Luciferase; B-Actin: Beta-Actin; Control: Vector control tumor cell; KO: CEP55 KO).

## Data Availability

Data is available on request from the corresponding author due to privacy.

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
