# Peer review of "Centrosomal Protein 55 (CEP55) Drives Immune Exclusion and Resistance to Immune Checkpoint Inhibitors in Colorectal Cancer"

_vaccines, 2024, doi:10.3390/vaccines12010063_

Round 1

Reviewer 1 Report

Comments and Suggestions for Authors

The investigation of role of CEP55 in ICI resistance in colorectal cancer by Wangmo et al. is interesting. However, study is incomplete without addition of few more mice in the orthotopic mouse to conclude what has been claimed.

Beside that I have few more minor comments on the study to improvise the manuscript.

1) Detailed statistical analysis is needed, all t- tests are unpaired? be specific in each figures.

2) Figure 1 and subsequent figures- mention the type of graph presented in the figure caption. Figure 1D- graph looks like there is no significant changes has been observed in CD4+ infiltration however it has been shown as significant difference. 

3) Figure- 2J- Segregate cell lines first as human and mouse and then subdivide into MSI and MSS. Why MC38 MSI cell  line has similar or over-expression of CEP55 compared to MSS CT26 cell line?

Comments on the Quality of English Language

Minor English editing is required.

Author Response

Response to the Reviewers:

We express our gratitude to the reviewers for their valuable feedback and insightful comments. In response to each comment, we conducted additional experiments and/or revised the in-text language to reflect and incorporate the suggested improvements.

Reviewer 1:

Comment: The investigation of role of CEP55 in ICI resistance in colorectal cancer by Wangmo et al. is interesting. However, the study is incomplete without addition of few more mice in the orthotopic mouse to conclude what has been claimed.

Response: We thank the reviewer for acknowledging the interest in our study on CEP55 in ICI resistance in colorectal cancer. We concur with the reviewer's observation that the study would benefit from the inclusion of more mice in the orthotopic animal experiment to establish a conclusive link between CEP55 knockout and an improved response to ICIs. Unfortunately, due to time constraints in the revision process, we were unable to incorporate additional animals into this manuscript.

To address this limitation, we have modified the language in the discussion sections to convey a more cautious interpretation of our results. The revised statement reads, " This investigation highlights a potential regulatory role for CEP55 in the context of CRC. Our initial orthotopic model study, depicted in Figure 5, explores the response of CEP55 knockout tumors to ICIs. While we observed a reduction in endpoint tumor volumes in a small sample size of mice, it is crucial to acknowledge the limitations of this investigative analysis. Future studies are warranted to augment the sample size, employing an appropriate power analysis to discern whether the observed tumor volume reduction is statistically and biologically significant. These initial observations, though intriguing, necessitate rigorous mechanistic validation, and increasing the sample size will be imperative for deriving translationally viable insights from the study."

Comment: Detailed statistical analysis is needed, all t- tests are unpaired? Be specific in each figure.

Response: We have updated each figure to incorporate its respective statistical analysis. The comparison of the two groups utilized the unpaired Student’s t-test unless otherwise specified. In instances of multiple group comparisons, we employed a one-way analysis of variance (ANOVA) followed by Bonferroni correction. A detailed description of the statistical analysis methods can be found in the Method section. A significance threshold of p-value < 0.05 was considered statistically significant.

Comment: Figure 1 and subsequent figures- mention the type of graph presented in the figure caption. Figure 1D- graph looks like there are no significant changes observed in CD4+ infiltration; however, it has been shown to have a significant difference. 

Response: In this data analysis, we stratified TCGA samples into two groups based on their CEP55 expression levels: 1) the upper 1/3 expressors of CEP55 and 2) the lower 1/3 expressors of CEP55. Subsequent to this stratification, we employed CIBERSORT to estimate the relative abundance of CD4 T cell infiltrate. The two groups' relative abundances of CD4 T cells were compared using the unpaired t-test. A significance threshold of p-value < 0.05 was considered statistically significant.

Comment: Figure- 2J- Segregate cell lines first as human and mouse and then subdivide into MSI and MSS. Why MC38 MSI cell line has similar or over-expression of CEP55 compared to MSS CT26 cell line?

Response: The cell lines are stratified by human and mouse, followed by MSI and MSS status. Notably, while the MC38 cell line is classified as an MSI subtype, studies reveal dynamic changes in its tumor microenvironment when implanted in mice. Late-stage MC38 tumors exhibit features resembling human CRC tumors, including T cell exclusion and T cell exhaustion—critical aspects of human CRC-MSS tumors (Shields et al.).

Despite microsatellite instability status serving as a benchmark for tumor treatment, the response of tumors to immune cells and treatments such as immune checkpoint inhibitors involves a complex interplay of multiple factors. Although, MC38 and CT26 are commonly used mouse CRC cell lines, the disparities in tumor growth conditions between in vitro cell culture and the human colon prompt further investigation. Future studies will investigate the mechanisms underlying the interplay between CEP55 and microsatellite instability status in human versus mouse models.

Shields, N. J., Peyroux, E. M., Ferguson, A. L., Steain, M., Neumann, S., & Young, S. L. (2023). Late-stage MC38 tumors recapitulate features of human colorectal cancer–implications for appropriate timepoint selection in preclinical studies. Frontiers in Immunology14, 1152035.

Reviewer 2 Report

Comments and Suggestions for Authors

This manuscript describes very well the role of CEP55 in CRC and its potential to be used as a target to improve the efficacy of ICIs in patients with MSS CRC.

Although the rationale of the study and experiments are presented in a good manner there are several aspects that were not explored by authors in the manuscript:

1) Authors analyzed the expression of CEP55 in both mouse and human CRC cell lines, but then only mouse cell lines were used to perform CEP55 KO and to study the tumor microenvironment in tumor implanted in mice. Why authors did not perform CEP55 KO, organoid experiment and in vivo studies on mice using also human CRC cell lines? Sometimes the tumor microenvironment established by human derived tumor could be different from the tumor microenvironment established by mouse derived cell lines. Comparing experiments using mouse and human derived CRC cell lines would have given to the reader two important information: A) If there is any difference between human and mouse CRC biology after CEP55 KO, especially at level of TME. B) It would make results presented stronger and more solid.

2) One of the main mechanisms that cancer cells use to induce T-cells exhaustion and Treg accumulation in the TME is the production and release of inhibitory cytokines. Authors claimed that CEP55 KO was correlated to an enhanced infiltration and activation of T cells and a concomitant reduction of Tregs in the TME. Did authors evaluated if CEP55 KO tumors had also a reduction or inhibition of production of inhibitory cytokines compared to tumors expressing CEP55?

3) Authors showed that CEP55 KO tumors became more sensitive to the activity of ICIs targeting PD-1. What about PDL-1 expression on cancer cells? Could CEP55 KO led to a reduction of PDL-1 expression on cancer cells, rendering CEP55 KO tumors also more sensitive to the activity of anti-PDL-1 ICIs?

Author Response

Reviewer 2:

Comment: This manuscript describes very well the role of CEP55 in CRC and its potential to be used as a target to improve the efficacy of ICIs in patients with MSS CRC. Although the rationale of the study and experiments are presented in a good manner, there are several aspects that were not explored by the authors in the manuscript.

Response: We thank the reviewer for complementing the rationale and the experimental execution of our study. We have addressed all comments below.

Comment: Authors analyzed the expression of CEP55 in both mouse and human CRC cell lines, but then only mouse cell lines were used to perform CEP55 KO and to study the tumor microenvironment in tumors implanted in mice. Why authors did not perform CEP55 KO, organoid experiments and in vivo studies on mice using also human CRC cell lines? Sometimes the tumor microenvironment established by human-derived tumor could differ from the tumor microenvironment established by mouse-derived cell lines. Comparing experiments using mouse and human-derived CRC cell lines would have given to the reader two important information: A) If there is any difference between human and mouse CRC biology after CEP55 KO, especially at the level of TME. B) It would make the results presented stronger and more solid.

Response: 1.) We endeavored to suppress CEP55 expression in our organoid model system, a task filled with experimental and technical challenges. During the clonal selection period, achieving a reduction in CEP55 expression proved limited, reaching at most ~30%, and was predominantly transient. This transient reduction would be overshadowed by the outgrowth of clones expressing CEP55. We utilized cell lines lacking CEP55 to overcome this hurdle, providing a more manageable and manipulable model system.

2.) Our choice not to employ human CRC cell lines for CEP55 knockout (KO) stems from the specific focus of our study on unraveling the immunomodulatory role of CEP55 within the context of CRC. To ensure the maintenance of a syngeneic and immunocompetent model system, we opted for CT26 cells derived from Balb/c mice and MC38 cells derived from C57BL/6 animals. This strategic approach provided a background conducive to the study of immune regulation.

Comment: One of the main mechanisms cancer cells use to induce T-cells exhaustion and Treg accumulation in the TME is the production and release of inhibitory cytokines. Authors claimed that CEP55 KO was correlated to an enhanced infiltration and activation of T cells and a concomitant reduction of Tregs in the TME. Did authors evaluate if CEP55 KO tumors also had a reduction or inhibition of the production of inhibitory cytokines compared to tumors expressing CEP55?

Response: In this study, we did not collect peripheral blood, plasma, or serum at the experiment endpoint to assess changes in pro-inflammatory or anti-inflammatory cytokines. In the discussion section, we have acknowledged this limitation and highlighted that future mechanistic experiments will be conducted to explore how the presence of CEP55 influences the expression and abundance of cytokines. This approach aims to provide a more comprehensive and mechanistic understanding of CEP55-dependent T cell regulation.

Comment: Authors showed that CEP55 KO tumors became more sensitive to the activity of ICIs targeting PD-1. What about PDL-1 expression on cancer cells? Could CEP55 KO led to a reduction of PDL-1 expression on cancer cells, rendering CEP55 KO tumors also more sensitive to the activity of anti-PDL-1 ICIs?

Response: We appreciate the reviewer's insightful comment and would like to provide additional context. The observation regarding sensitivity to ICIs was derived from a small sample size (n=3 per group) in our experiment. Recognizing the importance of strengthening statistical power, future experiments are planned to increase the sample size, enabling more robust and conclusive findings.

Addressing the question of PD-L1 expression changes in CT26 CEP55 KO and CT26 CEP55 WT; we performed immunofluorescence on FFPE tissues using a PD-L1 antibody. The data did not reveal a significant difference in PD-L1 expression between CT26 CEP55 KO and CT26 CEP55 WT tumors. This information, included in the supplemental material, suggests that how CEP55 regulates response to ICIs might involve intrinsic mechanisms independent of a cancer cell or stromal cell expression of PD-L1. Future experiments will focus on unraveling the more mechanistic aspects of CEP55's immunoregulatory function.

The revised text includes the following statement: “Our investigation aimed to determine whether the knockout of CEP55 in cancer cells could potentially elevate the expression of PD-L1, thereby enhancing sensitivity to anti-PD1 therapy. Immunofluorescence analysis was conducted on orthotopic tumor tissues from CEP55 knockout and CT26 wild-type (WT) tumors. Surprisingly, we did not observe a significant increase in PD-L1 expression in CEP55 knockout tumors compared to CT26 WT tumors (p=0.2998) (Supplemental Figure 1). This suggests that CEP55 may influence sensitivity to immune checkpoint inhibitors (ICIs) through alternative intrinsic immune regulatory mechanisms. Our results suggested that treating CEP55 knockout tumors with anti-PD1 reduces tumor growth and size, suggesting the potential for improved efficacy by inhibiting CEP55. It is crucial to acknowledge that the sample size for these observations is limited, emphasizing the need for future investigations to characterize the synergistic effects of CEP55 knockout tumors more comprehensively. A larger sample size and robust mechanistic validation will be integral in further elucidating the impact of CEP55 inhibition on the efficacy of ICIs in tumors.

Round 2

Reviewer 1 Report

Comments and Suggestions for Authors

Incorporate the responses to comments in the final manuscript at appropriate places without hampering flow of readability. Manuscript is ready to accept after this revision.

Author Response

As recommended, we have incorporated the suggested changes into our revised manuscript.

Reviewer 2 Report

Comments and Suggestions for Authors

I thanks authors for their valuable responses to my comments.

In my opinion there are a couple of things that authors need to add in the discussion as limitations of this study:

1) add a sentence like that:

One of the main mechanisms that cancer cells use to induce T-cells exhaustion and Treg accumulation in the TME is the production and release of inhibitory cytokines. In this study, we did not collect peripheral blood, plasma, or serum at the experiment endpoint to assess changes in pro-inflammatory or anti-inflammatory cytokines.  Future mechanistic experiments will be conducted to explore how the presence of CEP55 influences the expression and abundance of cytokines involved in the T-cells exhaustion and Treg accumulation in the TME.

2) Authors need to add a sentence stating that this study provide data and conclusion valid in pre-clinical mice models. This data cannot be translated to predict the effect of knocking out CEP55 in humans. Only further studies conducted in humans can fully elucidate the importance of this target for cancer research and treatment in humans.  

Author Response

Thank you for offering constructive feedback on our revised manuscript. In response to your suggestions, we have incorporated the following paragraph into the limitation section of our study.

Inhibitory cytokine production is crucial for inducing T cell exhaustion and Treg accumulation in the TME. However, this study did not assess changes in pro-inflammatory or anti-inflammatory cytokines. Additional mechanistic experiments are required to understand how CEP55 influences the expression and abundance of inhibitory cytokines. We acknowledge that the conclusions drawn from this study apply exclusively to pre-clinical mouse models and may not predict similar outcomes in humans.